# Study on the Mechanical and Leaching Characteristics of Permeable Reactive Barrier Waste Solidified by Cement-Based Materials

**DOI:** 10.3390/ma14226985

**Published:** 2021-11-18

**Authors:** Xuan Chen, Wenkui Feng, Hao Wen, Wei Duan, Chongxian Suo, Mingxing Xie, Xiaoqiang Dong

**Affiliations:** 1College of Civil Engineering, Taiyuan University of Technology, Taiyuan 030024, China; c13546341719@163.com (X.C.); f15513682175@163.com (W.F.); duanwei@tyut.edu.cn (W.D.); Suochongxian0053@link.tyut.edu.cn (C.S.); star7822378@163.com (M.X.); 2School of Civil Engineering, Southwest Jiaotong University, Chengdu 610031, China; wenha0@yeah.net

**Keywords:** PRB waste, wet-dry cycle, unconfined compressive strength, initial resistivity, leaching characteristics, microstructural characteristics

## Abstract

The durability against wet-dry (w-d) cycles is an important parameter for the service life design of solidified permeable reactive barrier (PRB) waste. This study introduces the potential use of cement, fly ash, and carbide slag (CFC) for the stabilization/solidification (S/S) of PRB waste. In this study, solidified PRB waste was subjected to different w-d cycles ranging in times from 0 to 10. By analyzing the mass loss, the unconfined compressive strength (UCS), initial resistivity (IR), and the ${\mathrm{Mn}}^{2+}$ leaching concentration under different durability conditions, the results demonstrate that these variables increased and then tended to decrease with the number of w-d cycles. The UCS of contaminated soil is significantly correlated with IR. Moreover, scanning electron microscopy (SEM), energy dispersive spectroscopy (EDS), and X-ray diffraction (XRD) analyses indicate that the hydration products calcium silicate hydrate (C-S-H) and ettringite (AFt) are the main reasons for the enhancement of the UCS. However, the increase in ${\mathrm{Mn}}^{2+}$ concentration leads to a decrease in hydration products and the compactness of solidified soil, which has negative effects for the UCS and the leaching ion concentration. In general, the durability exhibited by the PRB waste treated with S/S in this paper was satisfactory. This study can provide theoretical guidance for practical engineering applications.

## 1. Introduction

Acid mine drainage (AMD) is caused by the oxidation of sulfide minerals after being exposed to oxygen and water, which remains a serious environmental challenge for the mining industry [1,2]. Manganese is a major contaminant in AMD. Its toxicity is correlated with kidney, lung, and intestinal damage, and its chemical compounds can legitimately be predicted to be carcinogenic [3]. Numerous studies have been conducted by various experts and scholars on the safe disposal of manganese associated with acid mine drainage [2,4,5]. Permeable reactive barrier (PRB) technology is the most visible and influential [6,7]. PRBs can be used as removable, semipermanent, or permanent devices [8]. PRBs can eliminate heavy metal ions found in acid mine drainage to a tune of more than 95% [9,10]. PRB reaction materials are generally carbon/zero-valent iron (ZVI combination) [11], fly ash [12], red mud [13,14], or a mixture of various materials [15,16]. However, due to the active material’s reactivity, durability, and site limits, the active material must be changed regularly. Additionally, because the heavy metal ions in PRB waste are in high quantities, removing the replacement material constitutes a new management concern.

In recent years, solidification/stabilization technology (S/S technology) has been more popular as a proven remediation solution for polluted sites and solid waste landfills [17]. Ordinary Portland cement is commonly employed as the fundamental cementitious material component in S/S technology because of its great strength, durability, and availability [18]. However, because the cement production process accounts for 5–10% of worldwide anthropogenic emissions, the cement industry must produce environmentally friendly, low-carbon, and effective cementitious materials for soil restoration in new and sustainable ways [19,20]. As a result, in the soil consolidation process, alternative materials such as fly ash, blast furnace slag, geopolymers, and activated magnesium oxide are utilized as partial or total alternatives for silicate cement [21]. When mixed with cement, these alkaline solid wastes can increase one or both of the mixture’s strength and durability qualities [22] and their ability to effectively sequester. The w-d cycle can significantly impact the mechanical behavior of soils and their performance in a variety of geotechnical applications, including foundations, pavements, embankments, and engineered barriers in waste management systems [23,24]. Fly ash, a readily available industrial waste, can be employed as a suitable cementitious material additive in this context. Liao et al. [25] investigated the mechanical properties of cementitious materials exposed to the w-d cycle and discovered that cementitious materials’ compressive strength, mass loss, and dynamic elastic modulus rose first and then declined. Wang et al. [26] investigated the hardening of cementitious materials using MgO-fly ash mixes. According to Wei et al. [27], the addition of fly ash to cellulose fiber concrete modifies the initial pore structure and increases its durability against sulfate. Wdah et al. [28] investigated the effects of w-d cycles on red mud waste solidified with desulfurization gypsum-fly ash. The effect of the cycle was investigated, and the practicality of employing resistivity response stress was confirmed. Du et al. [29] studied the effect of w-d cycles on the properties of calcium carbide slag-MgO stabilized kaolin and found that the UCS of cured soil decreased with increasing w-d cycles. Kampala et al. [30] studied the durability properties of calcium carbide slag and fly ash solidified silt under the action of w-d cycles and found that the mixture could be used for soil curing to obtain medium strength and better durability geological material. These earlier studies demonstrated that curing of contaminated waste with different composite solidified materials under the influence of w-d cycles showed good durability. However, very limited research has addressed the impacts of w-d cycles on the durability of composite materials solidified PRB wastes.

Accordingly, the objective of this study is to investigate the effect of w-d cycle on the durability of CFC solidified PRB waste. This study is based on a nonrecyclable PRB waste, consisting mainly of a 7:3 mixture of red mud and loess [31], solidified with CFC to investigate the mechanical and leaching properties under the influence of w-d cycles, and microscope analysis by scanning electron microscopy (SEM), energy dispersive spectroscopy (EDS), and X-ray diffraction (XRD). This research can provide feasible solutions for the use of PRB solid waste in practical engineering.

## 2. Materials and Methods

### 2.1. Materials

Acidic manganese-contaminated PRB waste (RLC) consisted of 7:3 red mud and loess. (The results of the study demonstrated that red mud and loess at 7:3 can be used as PRB material to adsorb more than 95% of heavy metal ions, as detailed in [31].) The initial moisture content of the RLC is 60%. The sieved RLC was placed in sieves with particle sizes less than 2 mm and combined with the acidic solution. The basic physical parameters of RLC were obtained by compaction tests according to standard for geotechnical testing method (GB/T 50123-2019), as shown in Table 1.

The RCL was formulated in the laboratory. The red mud used in this study (Figure 1a) was obtained from the Liulin aluminum plant in Taiyuan, Shanxi Province, China. The red mud employed in this work is a highly alkaline leftover from the Bayer process [32], which extracts alumina from produced bauxite, accounting for more than half of ${\mathrm{SiO}}_{2}$ and ${\mathrm{Al}}_{2}O_{3}$. The loess has a yellow appearance. Its plastic limit is 14.55%, the liquid limit value is 25.03%, and the plasticity index is 10.48%, which is typical of reconstituted loess from Shanxi (Figure 1b). According to Figure 2, the uniformity coefficients for red mud and loess were 6.68 and 8.32, respectively, and the curvature coefficients of red mud and loess were 1.401 and 1.648, respectively.

The ordinary Portland cement (Figure 1c) used in this study had a strength grade of 42.5 and came from the Taiyuan Lionhead Cement Plant. Fly ash (Figure 1d) was collected from Taiyuan No. 1. The thermal power plant, Taiyuan, Shanxi, China, contained large amounts of ${\mathrm{SiO}}_{2}$ and ${\mathrm{Al}}_{2}O_{3}$, which were used as partial cement replacements in this study. Carbide slag (Figure 1e) was collected from Yushe Chemical Co., Ltd, Taiyuan, Shanxi Province, China. and was beige, with CaO as the main chemical composition. The chemical compositions of the three hardened materials are shown in Table 2.

### 2.2. Specimen Preparation

First, the pH of the water sample was adjusted to approximately 3.0 using a dilute solution, and then two concentrations of ${\mathrm{Mn}}^{2+}$(chemical reagent of choice ${\mathrm{MnSO}}_{4}\cdot H_{2}O$, analytical purity purchased by Tianjin Tianli Chemical Reagent Co, China) at 1000 $\mathrm{mg}\cdot L^{-1}$ and 5000 $\mathrm{mg}\cdot L^{-1}$ were added to simulate acidic mine drainage. A mixture of red mud and loess was placed in a soil column, and the RLC was removed after 30 days of soaking with simulated acid mine drainage, representing the waste at both pollutant concentrations. The RLC was dried at 60 °C, crushed, and passed through a 2mm sieve; cement, fly ash, and carbide slag were mixed at 8%, 8%, and 5% (results were obtained by orthogonal tests in the previous period) of the dry weight of RLC, respectively, and the optimum moisture content of 33% and maximum dry density of 1.47 $g\cdot {\mathrm{cm}}^{-3}$ were obtained by compaction tests. The mixture was mixed to an optimum moisture content of 33%, and the specimens were prepared by the hydrostatic method to produce a dry density of 95% of the maximum dry density. The specimen was a cylinder of 50 mm in diameter and 50 mm in height. Cement-fly ash-carbide slag solidified in RLC is defined as FCCR1 and FCCR5. Three parallel samples were prepared for each group and placed in a curing chamber at a temperature of 20 °C ± 2 °C and relative humidity of 95% for 28 and 60 days, respectively.

Specimens for microscopic testing were immersed in alcohol to stop the hydration process [32], air-dried, and removed using 1 cm × 1 cm × 0.3 cm fine-grained sandpaper for SEM and XRD testing.

### 2.3. Testing Procedures

#### 2.3.1. W-D Cycle Test

The specimens were tested for w-d cycles according to ASTM D4843-88 [33].

Step 1: Weigh the samples before the test.

Step 2: Place the samples into the oven at a temperature of 70 °C for 12 h, then take the sample from the oven and weigh it.

Step 3: Place and submerge the samples in distilled water for 24 h.

Step 4: Weigh the samples after removing them from distilled water.

The number of w-d cycles designed was 0, 1, 3, 5, 7, and 10. The mechanical and leaching characteristics and microstructural characteristics of the FCCR were analyzed at different cycle times.

#### 2.3.2. UCS Test

The UCS test was performed on the test procedure for inorganic bonding materials for road engineering (JTG/T E51-2009) [34], which was conducted using an electonic universal testing machine. Subsequently, these samples were pressed at a rate of 1 mm/min until destruction.

#### 2.3.3. Resistivity Test

The resistivity test was performed using a digital bridge (TH2828A) to record the IR as well as the process resistivity change [32]. The w-d test does not damage the sample, and the completed w-d test sample can be placed in the press for resistivity testing. After each w-d cycle, the sample surface was dried of moisture. Graphite was evenly applied to the surface of both ends of the sample, copper electrode pads were placed, and the leads were connected to a TH2828A-type LCR digital bridge for testing.

#### 2.3.4. Leaching Concentration Test

The leaching toxicity test was performed using the sulfuric acid and nitric acid method of the solid waste leaching toxicity extraction method (HJ/T299-2007) [35]. The concentration of pollutants in the filtrate was determined by inductively coupled plasma emission spectrometer.

#### 2.3.5. Microscopic Testing

XRD analysis was conducted on an FCCR to judge the formation of the new phase, and the natural drying samples were scanned with 2ⱷ ranging between 10° and 70°. The specimens’ pore structure and surface morphology were then observed at 5000×, 10,000×, and 20,000×. Moreover, SEM and EDS analysis were conducted using Hitachi TM3000 scanning electron microscope, Japan. The test flow is shown in Figure 3, and the experimental design is shown in Table 3.

## 3. Results and Discussion

### 3.1. UCS

The influence of the w-d cycle on the UCS of the specimens under various durability circumstances is shown in Figure 4. It is evident from the figure that the strength of the FCCR tends to rise with the number of w-d cycles, reaching a peak at the seventh w-d cycle. The inclusion of fly ash and carbide slag increased the specimen strength at first, but when the w-d cycle took effect, the time to achieve the intensity peak shifted back. The high-temperature environment encourages fly ash and carbide slag to enhance the hydration rate during the dry cycle [36]. In contrast, the aqueous environment favors the combination of fly ash and carbide slag to increase the hydration rate during the wet cycle [37]. At the same time, fly ash has a “filling effect” that can limit the soil’s pore space, increasing its strength. This shows that fly ash changes the pore structure in solidified soil, resulting in decreased soil porosity and increased soil structure compactness. The influence of the early w-d cycle on solidified samples was reduced [38,39].

The samples’ strength deteriorated after the seventh w-d cycle, notably as the ${\mathrm{Mn}}^{2+}$ content climbed from 1000 $\mathrm{mg}\cdot L^{-1}$ to 5000 $\mathrm{mg}\cdot L^{-1}$. At 28 and 60 days, the maximal strength of the FCCR fell by 2.3% and 3.5%, respectively. The impact of manganese ions on FCCR strength is mainly due to a high concentration of ions slowing down the hydration process, lowering the hydration products and thus lowering the strength. The damage to the soil structure worsens from physical erosion as the number of w-d cycles rises, and cracks emerge in the pore structure and spread, leading to structural flaws and a fall in FCCR.

### 3.2. IR

Figure 5 depicts the IR trend as the number of w-d cycles increases. The IR increases with the number of w-d cycles. As seen in the graph, it then decreases, which is compatible with the law of change of UCS. The first seven w-d cycles show a rise in IR, and after the seventh w-d cycle, the IR displays a declining trend. This is because the pore space of the soil body expands, and cracks form throughout the w-d cycles. This is mainly because the soil generates fissures during the w-d cycle, and pore size becomes more prominent. Additionally, to a certain extent, it will block some of the conductive paths between soil particles, and the soil particle orientation is weakened [40]. Thus, the conductivity between soil particles is reduced, and the resistivity shows an increasing trend.

From the 7th w-d cycle, the degree on the soil gradually increases, the soil particle skeleton is compressed, the soil is damaged, the larger pores between particles are compressed into tiny pores, the pore water saturation increases, and the pore water conductivity is enhanced; thus, the resistivity shows an attenuation trend. As the pore water saturation rises and the pore water’s electrical conductivity rises, the resistance falls. Because manganese ions are innately conductive, they affect FCCR resistivity. The stronger the conductivity and lower the resistivity of the soil, the higher the ion concentration in the pore solution, which is also consistent with the trend in UCS with ion concentration.

### 3.3. Correlation of UCS and IR

Figure 6 depicts the connection between the UCS and IR for various w-d cycles. It can be seen that the dark area near the fitted curve of UCS and IR is the 95% confidence interval of UCS, which includes more than half of the test data points in the 95% confidence interval, and includes basically all of the test data points in the whole 95% prediction interval, indicating that the test of IR of the test block can have 95% prediction to predict the UCS of the test block. The regression equations for the four fitted curves are listed in the Table 4. It can be deduced from the table and graphs that:

(a) At the same ${\mathrm{Mn}}^{2+}$ concentration, the resistance increases with curing time and IR.

(b) For a given IR, the higher the ${\mathrm{Mn}}^{2+}$ concentration is, the greater the compressive strength.

(c) For the same resistivity increment, the slope of the straight line increases with increasing ${\mathrm{Mn}}^{2+}$ concentration, showing that the compressive strength increases with increasing ${\mathrm{Mn}}^{2+}$ concentration.

Many researchers have studied and established the relationship between the UCS and IR of solidified soils [41,42,43,44,45] who discovered that the UCS and IR had a good linear relationship. The results of this paper are consistent with the above researchers. As a result, a resistivity method can be a valuable tool for assessing the quality of hardened soils and can be applied to various engineering tests.

### 3.4. Leaching Concentration

Figure 7 depicts the increase in ${\mathrm{Mn}}^{2+}$ leaching concentration as the number of w-d cycles increases. The leaching concentration of FCCR increases and then stabilizes as the number of w-d cycles increases, but all of these values are below the primary effluent discharge standard of 2.0 $\mathrm{mg}\cdot L^{-1}$, indicating that they will not harm the production and living environment or the human body [43]. The concentration of leached ions can detect a noticeable rise between zero and three cycles of the w-d cycle. This is because each w-d cycle develops small cracks on the surface and interior of the sample [46], resulting in increased ion leaching and FCCR mass loss. When the sample undergoes three w-d cycles, the leached concentration tends to stabilize or even decrease: with the continuous hydration process, hydration ${\mathrm{Ca}(\mathrm{OH})}_{2}$ and calcium silicate hydrated (C-S-H) gelation products were generated, among which ${\mathrm{Ca}(\mathrm{OH})}_{2}$ and easily dissolved water-soluble ${\mathrm{Mn}}^{2+}$ formed precipitates. ${\mathrm{Ca}(\mathrm{OH})}_{2}$ was easily oxidized to the more stable tetravalent manganese oxide in the alkaline environment, and the hydration products C-S-H and AFt increased after the large consumption of ${\mathrm{Ca}(\mathrm{OH})}_{2}$ in the later phase. The system was mainly responsible for solidifying the continuously dissolved water-soluble ${\mathrm{Mn}}^{2+}$: hydration products’ physical adsorption and encapsulation. FCCR5-28 also had the largest concentration of leached ${\mathrm{Mn}}^{2+}$ and, as a result, the lowest UCS, implying that the initial ${\mathrm{Mn}}^{2+}$ concentration significantly impacts the FCCR’s w-d cycle endurance.

### 3.5. Mass Loss

As shown in Figure 8, capillary pores form on the surface of the FCCR, and tiny cracks appear; after ten w-d cycles, local detachment of the surface occurs, resulting in a mass change. The following two equations can calculate the mass change during w-d cycle action: Equation (1) is the mass loss for i w-d cycle actions, and Equation (2) is the cumulative mass loss for ten w-d cycle actions.
(1)$$\mathrm{ML}=\frac{\left(m_{0}-m_{i}\right)}{m_{0}}\;\times \;100\%$$
(2)$$\mathrm{CML}=\overset{10}{\underset{i=0}{\sum }}{\mathrm{ML}}_{i}$$
where $m_{i}$ is the mass of the specimen after i cycles and $m_{0}$ is the initial drying mass of the specimen.

Figure 9 represents the variation in the mass loss rate of FCCR with d-w cycles. From Figure 9a, it can be seen that the difference in the mass-loss rate per wetting cycle for FCCRs with two different pollutant concentrations at different maintenance ages is slight. During wet cycles 0–3, the mass-loss rate increased significantly with the action of the wet cycle. In the successive seven cycles, the mass loss rate of the specimens increased only slightly compared to the first three cycles and nearly stabilized. The specimen’s structure has not yet reached a dense condition, and the internal pores are numerous at the start of the cycle. The water evaporates quickly after three drying cycles, and the mass loss rate is considerable. Between cycles 3 and 10, the gelling material generated by FCCR hydration fills the interior pores. The mass loss continues at a nearly constant rate while compressive strength declines and the leached ion concentration decreases. This conclusion is consistent with the findings of Guo et al. [47].

As shown in Figure 9b, under the effect of w-d cycles, the cumulative mass loss of FCCR at higher concentrations was much more significant than that at lower concentrations. Compared with the initial mass, the cumulative mass losses in the last w-d cycle (No. 10) increased by 2.97% for FCCR1-28, FCCR5-28, FCCR1-60, and FCCR5-60. This phenomenon occurs because the high concentration of ${\mathrm{Mn}}^{2+}$ invades the surface pores of the sample, resulting in peeling of the sample skin and loosening of the internal structure, leading to significant quality changes. The strength and leaching ion concentration of the soil sample also changes significantly.

### 3.6. Microstructure Analysis

The extent of the effect of the w-d cycle on the hydration products of FCCR was further investigated by XRD analysis. Figure 10 displays the XRD graphs with various contaminant concentrations after 28 days of curing time under the effects of 0 and 10 w-d cycles. As can be observed in the figure, the comparison of the FCCR with the pure polluted waste identifies the development of new peaks, namely, the generation of calcium aluminosilicate hydrate (C-A-S-H) and ettringite (AFt) hydration products. The XRD plots for different numbers of w-d cycles show practically the same peaks, demonstrating that the effect of w-d cycles on the formation of gelling chemicals during the hydration process is negligible. The results of the analysis show that the hydration products of FCCR are mainly composed of calcium silicate hydrate (C-S-H), calcium aluminosilicate hydrate (C-A-S-H), and ettringite (AFt). In terms of XRD patterns, C-S-H gels are represented at the peaks of 2θ values corresponding to the vicinity of 32°, 42.5°, and 48.5°, which is consistent with existing studies [48,49]; also, C-A-S-H gels are likewise clearly observed near the 2θ values at 35.5°, 42.5°, and 45.5°, which is consistent with the results of existing studies [32,50]. The inclusion of fly ash and calcium carbide slag speeds up the hydration reaction process. It boosts the ionic activity, which helps to accelerate the volcanic ash reaction, carbonation reaction, and ion exchange reaction in the reaction system, resulting in more gelling material to strengthen the soil.

SEM images of specimens at 28 days of curing time were chosen. Figure 11a–d show SEM images at various magnifications without the w-d cycle, whereas Figure 10 and Figure 11e show SEM images at various contaminant concentrations with the w-d cycle. The atomic Ca/Si ratio of C-S-H is an important composition parameter that affects nonstructural characteristics of C-S-H (usually Ca/Si between 1.0 and 1.7 represent C-S-H) [51]. Referring to EDS component analysis results (see Figure 12), it is evident that a large amount of needle/rock-like calcified material formed in the soil samples after 28 days of curing time, that is, a massive amount of C-S-H/C-A-S-H and AFt was generated [51,52,53]. Through hydration, the glass beads of fly ash and needle-like ettringite are wrapped or bonded together by the flocculent C-S-H/C-A-S-H gel component, which binds the different forms together to form a denser structure, which is the main reason for providing soil strength. At room temperature, tricalcium silicate dicalcium silicate ($C_{3}S$) and dicalcium aluminate ($C_{2}S$) in the cement hydrate create C-S-H and ${\mathrm{Ca}(\mathrm{OH})}_{2}$. During the chemical reaction, fly ash is activated over time by ${\mathrm{Ca}(\mathrm{OH})}_{2}$ produced by the hydration of the cement and by ${\mathrm{Ca}(\mathrm{OH})}_{2}$ contained in the calcium carbide slag itself, and ${\mathrm{OH}}^{-}$ from the pore fluid breaks Si-O-Si and Al-O-Al bonds, the hydrated calcium silicate C-S-H and hydrated calcium aluminosilicate C-A-S-H with ${\mathrm{Ca}}^{2+}$ are generated with gelling activity, and the C-A-S-H is excited by ${\mathrm{OH}}^{-}$, ${\mathrm{SO}}_{4}{}^{2-}$, ${\mathrm{Ca}}^{2+}$ to generate the water-hardened AFt. Fly ash is primarily responsible for filling the microporous aggregates and reducing the water consumption during the reaction [54]. The flocculent C-S-H/C-A-S-H gels produced in FCCR can bind fine particles together and further promote the assembly of agglomerates. Therefore, the appearance of C-S-H/C-A-S-H gels is a key factor to improve the strength and durability of FCCR, especially for samples with long-term curing, which is also consistent with the study of [55].

Figure 11e,f show SEM images of different FCCR concentrations after ten w-d cycles. A comparison of Figure 11c,d shows that after ten w-d cycles, there is a significant reduction in calcarenite hydration products in the specimen, as well as a significant reduction in C-A-S-H/C-S-H flocculent colloidal material and a significant reduction in compactness. The pores between the particles become larger with the increasing of w-d cycles, which is the main reason for the variations in strength, indicating that the w-d cycle affected the soil to some extent [56]. From a geotechnical engineering perspective, FCCR has better durability against w-d cycling in addition to higher UCS values, which can be attributed to the increase in C-S-H/C-A-S-H and AFt during w-d. CFC can improve the short- and long-term UCS and durability of the fill material, which is beneficial for applications requiring practical engineering [57].

Figure 13 depicts a black-and-white binary image of the SEM processed with Image-Pro Plus, revealing which part of the soil has porosity. In the test block, porosity is defined as the ratio of the pore volume (area of the white zone in the black and white binary image) to the volume of the material (area of the black zone in the black and white binary image).

According to Table 5, for various FCCR concentrations, the porosity rose by 37.42% and 28.11% after ten w-d cycles, respectively. This is due to varying degrees of loosening of the soil skeleton in response to w-d cycles. The specimen’s hydration products formed are impacted and disrupted, creating a situation where the indicated pores become much more abundant. Although the porosity increases to varying degrees, the associated strengths are within the specification range, suggesting that FCCR is durable in both w-d cycles. This is also in line with the findings of the strength development program.

### 3.7. Discussion of the Mechanism

The fundamental mechanism of the FCCR hydration process is depicted in Figure 14. Cement in the dry state is mainly composed of tricalcium silicate dicalcium silicate ($C_{3}S$), dicalcium aluminate ($C_{2}S$), tricalcium aluminate ($C_{3}A$), and a small amount of sulfate (potassium salt, sodium salt) and gypsum (calcium sulfate dihydrate). During the hydration of cement, $C_{3}S$, $C_{2}S$, and $C_{3}A$ undergo complex hydration reactions with other components in cement to produce calcium alumina, i.e., ettringite AFt, calcium hydroxide, and C-S-H gel. $C_{3}S$ in the cement dissolves quickly in water, causing hydration, the first stage in the cement hydration reaction, which results in C-S-H gels and ${\mathrm{Ca}(\mathrm{OH})}_{2}$ crystals. Carbide slag adds ${\mathrm{OH}}^{-}$ to the system, breaking Si-O and Al-O bonds in the fly ash reactive body and providing the ${\mathrm{Ca}}^{2+}$ needed to make a hydraulic cementitious material [58,59] and make hydration products more stable and robust. ${\mathrm{SO}}_{4}{}^{2-}$ in solution is primarily utilized to increase the fly ash’s rate of active excitation and its degree of active excitation [19].

(a) The fly ash was stimulated by alkaline and sulfate interfaces, which formed hydrated calcium aluminate in addition to the C-S-H gel, and the fly ash’s ${\mathrm{Al}}_{2}O_{3}$ activity was effectively excited due to the coexistence of ${\mathrm{SiO}}_{4}{}^{4-}$ and ${\mathrm{Ca}(\mathrm{OH})}_{2}$.

(b)$\;{\mathrm{SO}}_{4}{}^{2-}$ reacts with ${\mathrm{Ca}}^{2+}$ and ${\mathrm{AlO}}_{2}{}^{-}$ trapped in the gel on the surface of the fly ash particles to create calcium bauxite, which is then released into the pore fluid. On the surface of the fly ash particles, calcium alumina creates a fibrous or cross-linked covering. The low density of this coating allows for easier ion dispersion and penetration, which boosts fly ash activity even more.

(c) ${\mathrm{SO}}_{4}{}^{2-}$ ions can replace some of the ${\mathrm{SiO}}_{4}{}^{4-}$ ions in the C-S-H gel, and the replaced ${\mathrm{SiO}}_{4}{}^{4-}$ ions are released and react with ${\mathrm{Ca}}^{2+}$ outside the inclusions to form the C-S-H gel again, allowing further excitation of the active fly ash; at the same time, the solubility of the active ${\mathrm{Al}}_{2}O_{3}$ increases significantly in the presence of ${\mathrm{SiO}}_{4}{}^{4-}$, promoting the excitation of the active ${\mathrm{Al}}_{2}O_{3}$.

In summary, the hydration reaction of FCCR has the following equation:

The equation for the hydration reaction of cement is:(3)$$\mathrm{CaO}+H_{2}O=\mathrm{Ca}{\left(\mathrm{OH}\right)}_{2}$$
(4)$$3\mathrm{CaO}\cdot {\mathrm{SiO}}_{2}+{\mathrm{nH}}_{2}O=\mathrm{xCaO}\cdot {\mathrm{SiO}}_{2}\cdot \left(n\;-\;3+x\right)H_{2}O+(3\;-\;x)\mathrm{Ca}{\left(\mathrm{OH}\right)}_{2}$$
(5)$$2\mathrm{CaO}\cdot {\mathrm{SiO}}_{2}+{\mathrm{nH}}_{2}O\cdot {\mathrm{SiO}}_{2}\cdot \left(n\;-\;2+x\right)H_{2}O+(2\;-\;x)\mathrm{Ca}{\left(\mathrm{OH}\right)}_{2}$$

The hydration reaction between cement and ${\mathrm{Ca}(\mathrm{OH})}_{2}$ contained in calcium carbide slag produces mainly C-S-H/C-A-S-H and Aft [30], thus continuously improving the strength of the cement system. Fly ash contains a large amount of reactive ${\mathrm{SiO}}_{2}$, ${\mathrm{Al}}_{2}O_{3}$, and other substances, and the hydration reaction with ${\mathrm{Ca}(\mathrm{OH})}_{2}$ produced by the hydration of cement and ${\mathrm{Ca}(\mathrm{OH})}_{2}$ contained in calcium carbide slag produces the following reaction equation:(6)$${\mathrm{SiO}}_{2}+\mathrm{nCa}{\left(\mathrm{OH}\right)}_{2}+{\mathrm{xH}}_{2}O\to \mathrm{nCaO}\cdot {\mathrm{SiO}}_{2}\cdot {\mathrm{xH}}_{2}O$$
(7)$${\mathrm{Al}}_{2}O_{3}+\mathrm{mCa}{\left(\mathrm{OH}\right)}_{2}+{\mathrm{yH}}_{2}O\to \mathrm{mCaO}\cdot {\mathrm{Al}}_{2}O_{3}\cdot {\mathrm{yH}}_{2}O$$
(8)$${\mathrm{Al}}_{2}O_{3}+3\mathrm{Ca}{\left(\mathrm{OH}\right)}_{2}+3({\mathrm{CaSO}}_{4}\cdot 2H_{2}O)+23H_{2}O=3\mathrm{CaO}\cdot {\mathrm{Al}}_{2}O_{3}\cdot 3{\mathrm{CaSO}}_{4}\cdot 32H_{2}O$$

With the processing of w-d cycles, the internal soil sample gradually generates tiny fissures under physical erosion: the pore space increases, and the wrapping force between soil particles decreases. The attached fine particles are scoured and reorganized under the action of dry and wet cycles. The soil strength gradually decays from the initial increase, corresponding to the surface appearance of the soil sample also causing the slag to fall off, leading to an increase in mass loss. In contrast, C-S-H/C-A-S-H and AFt act as wrapping adsorbents for ${\mathrm{Mn}}^{2+}$, and most of the ${\mathrm{Mn}}^{2+}$ can replace the aluminum in the structure of hydration products and then solidify stably in the soil. The leaching ion concentration is stabilized or even decayed. In general, the mechanical and leaching characteristics of FCCRs are affected to some extent by the w-d cycles. Still, they are all within the range of the specification so that FCCRs can provide good theoretical support for the application of practical projects.

## 4. Conclusions

This paper investigated the mechanical and leaching properties of FCCRs under w-d cycle conditions and SEM and XRD investigations of FCCRs. These results show that FCCR’s mechanical and leaching properties are stable in both dry and wet conditions. Its durability is outstanding; the proposed resistivity method effectively predicts the quality of solidified soil. It can be easily and quickly applied to practical engineering applications. The following conclusions can be drawn:

(1) There was a clear trend of increasing intensity in the UCS of FCCR during the first seven cycles and then the intensity decreased. This cementitious curing agent, under the influence of wet and dry cycles, shows excellent durability.

(2) The mass-loss rate of FCCR tended to grow during the first three w-d cycles and then stabilized, demonstrating that the w-d cycles had little effect on FCCR morphology. The high concentration of ${\mathrm{Mn}}^{2+}$ had an inhibitory or suppressive effect on the specimens.

(3) A good linear relationship was demonstrated between the UCS and IR after adjusting the UCS of the FCCR to the IR; therefore, the resistivity method can effectively evaluate the quality of solidified soil for engineering applications.

(4) XRD and SEM results indicated that the hydration products of FCCR are mainly composed of C-S-H/C-A-S-H gel and AFt gel. These gelling substances are adsorbed on the surface of red mud and fly ash and fill the pores of the specimen, and their strength and ion leaching concentration are not affected much even under the influence of w-d cycles.

## Figures and Tables

**Figure 1 materials-14-06985-f001:**
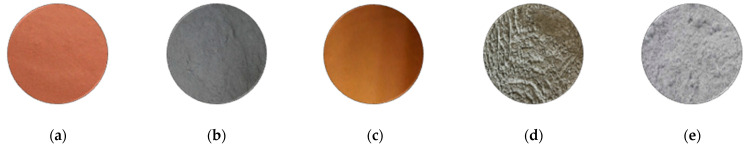
Test materials. (**a**) Red mud, (**b**) loess, (**c**) ordinary Portland cement, (**d**) fly ash, and (**e**) carbide slag.

**Figure 2 materials-14-06985-f002:**
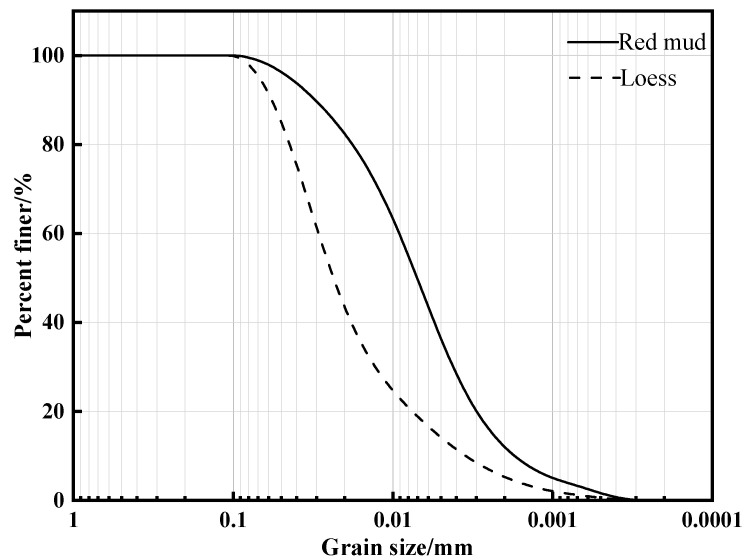
Particle size distribution of red mud and loess.

**Figure 3 materials-14-06985-f003:**
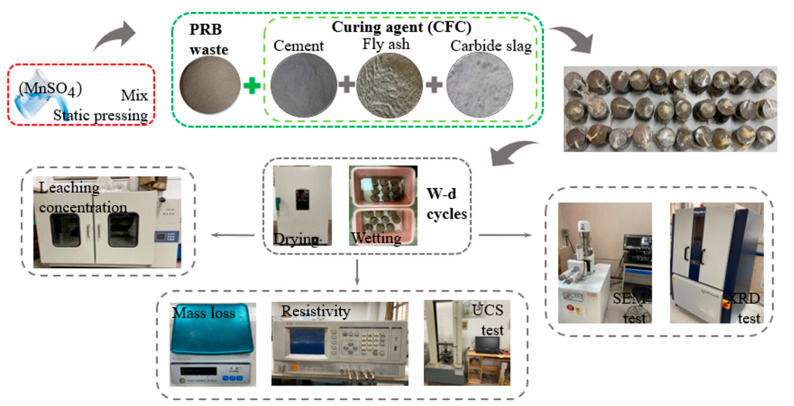
Testing process schematic.

**Figure 4 materials-14-06985-f004:**
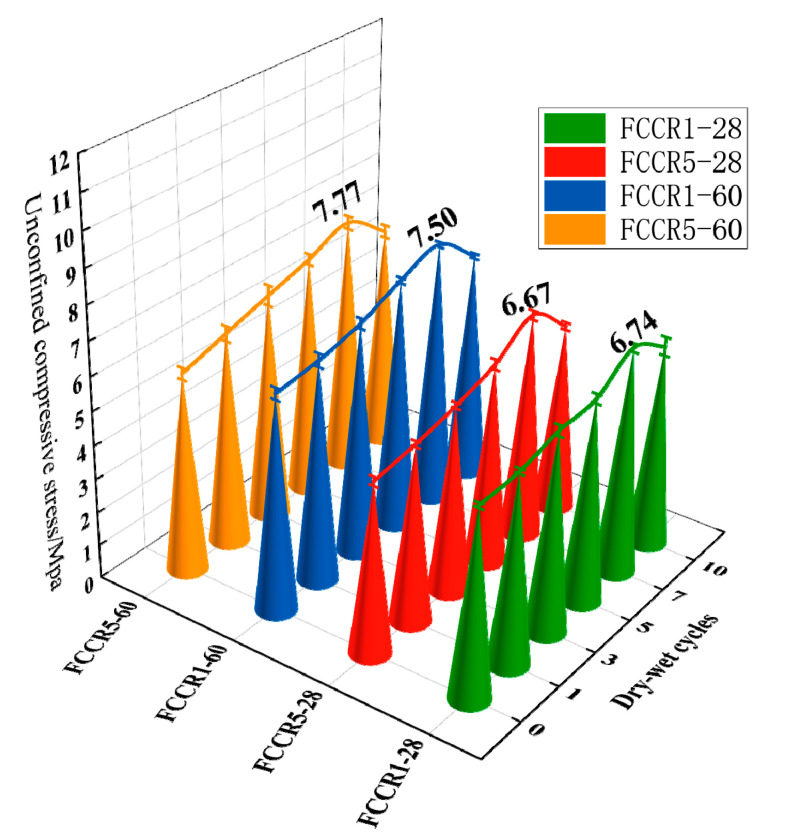
UCS of specimens under different durability conditions.

**Figure 5 materials-14-06985-f005:**
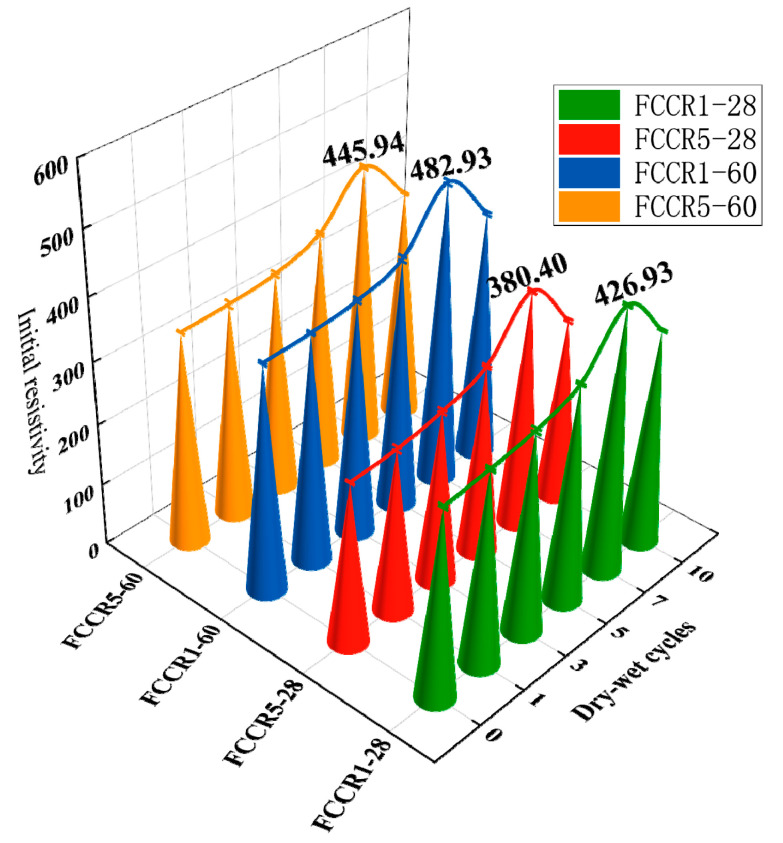
Curve of IR vs. different w-d cycles.

**Figure 6 materials-14-06985-f006:**
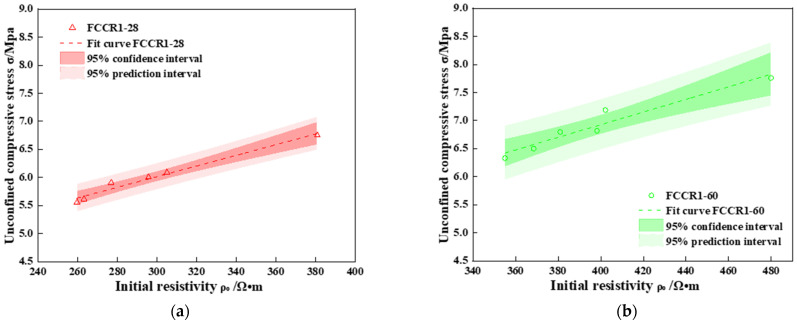
Relationship between UCS and IR. (**a**) UCS and IR fit curve of FCCR1-28; (**b**) UCS and IR fit curve of FCCR1-60; (**c**) UCS and IR fit curve of FCCR5-28; (**d**) UCS and IR fit curve of FCCR5-60.

**Figure 7 materials-14-06985-f007:**
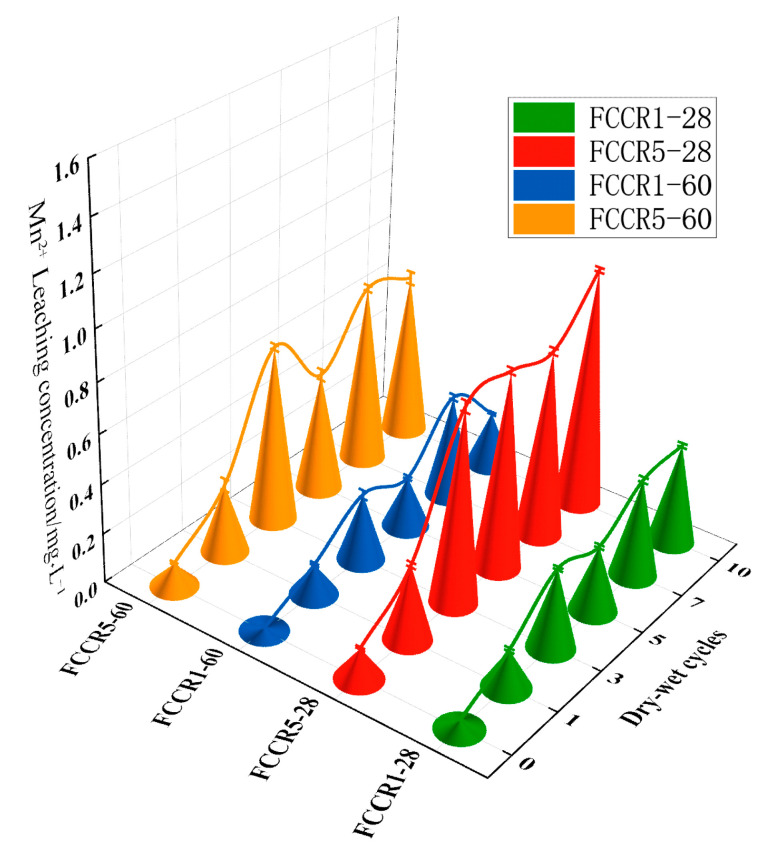
Leaching ${\mathrm{Mn}}^{2+}$ concentration under different durability conditions.

**Figure 8 materials-14-06985-f008:**
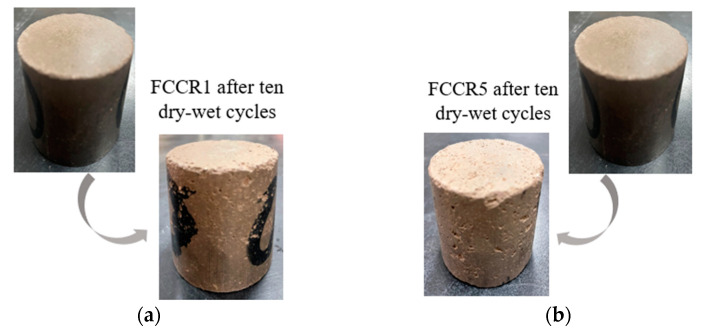
Surface features of FCCR after ten w-d cycles: (**a**) FCCR1 and (**b**) FCCR5.

**Figure 9 materials-14-06985-f009:**
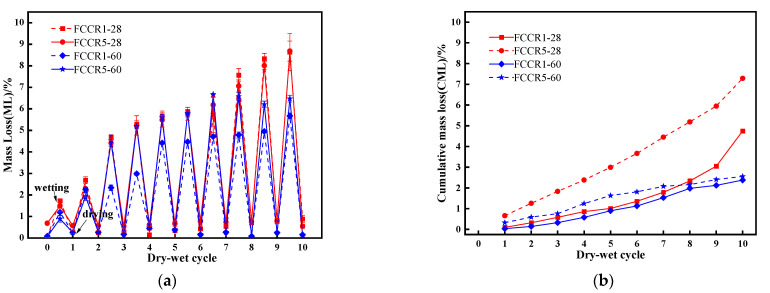
(**a**) Mass loss of specimens under different durability conditions; (**b**) cumulative mass loss of specimens under different durability conditions.

**Figure 10 materials-14-06985-f010:**
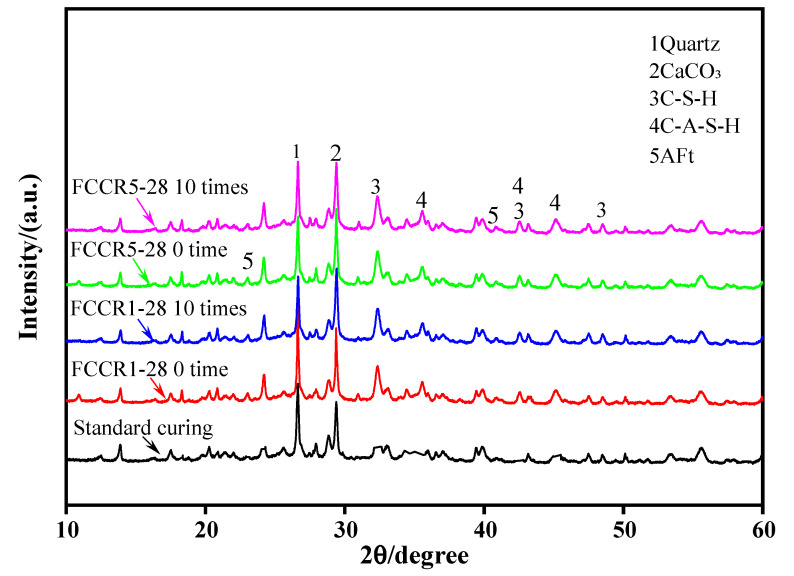
XRD pattern at 28 d of curing time under different durability conditions.

**Figure 11 materials-14-06985-f011:**
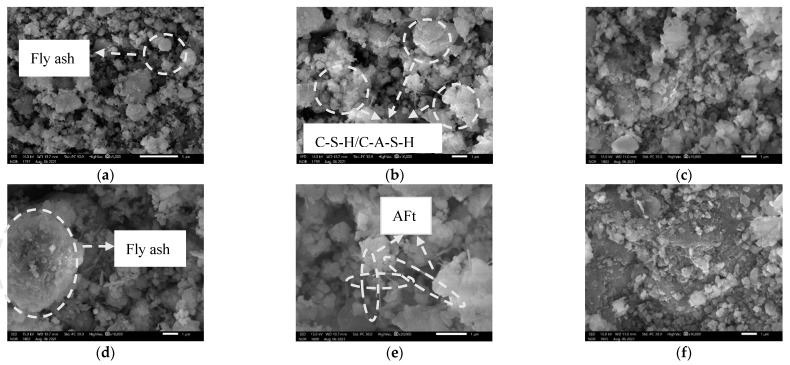
Microstructure after 28 d of curing: (**a**) × 5k; (**b**) × 10k; (**d**) × 10k; (**e**) × 20k. Microstructure after 10 w-d cycles: (**c**) FCCR1 × 10k; and (**f**) FCCR5 × 10k.

**Figure 12 materials-14-06985-f012:**
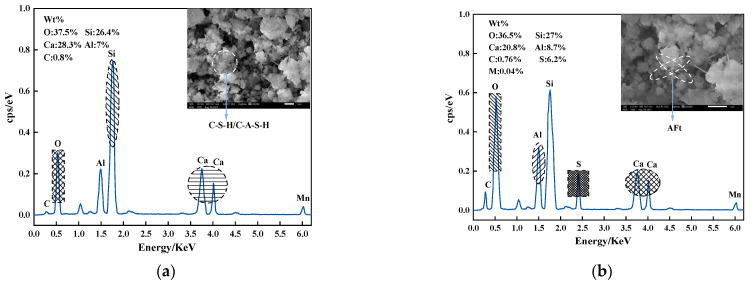
EDS images: (**a**) spectrum 1; (**b**) spectrum 2.

**Figure 13 materials-14-06985-f013:**
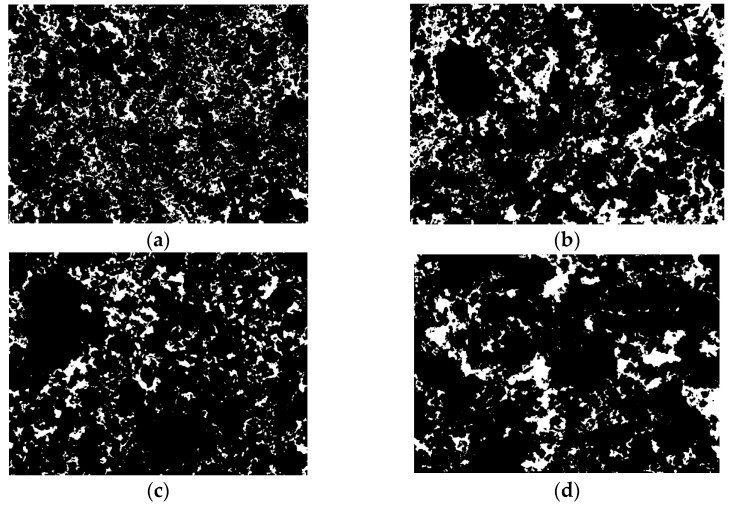
Black and white binary images of the specimens at 28 d of curing time: (**a**) RLC1-28 with 0 time; (**b**) RLC1-28 with 10 times; (**c**) RLC5-28 with 0 time; (**d**) RLC1-28 with 10 times.

**Figure 14 materials-14-06985-f014:**
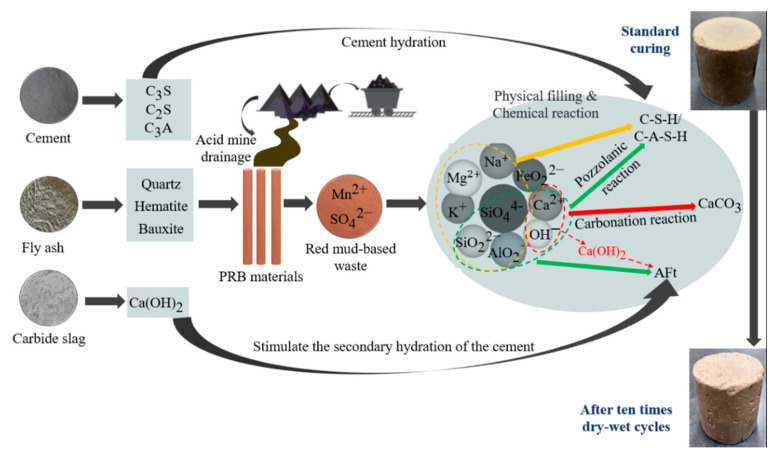
Schematic diagram of the mechanism.

**Table 1 materials-14-06985-t001:** Main physical properties of RLC.

Parameters	Values
Specific density	2.7
Liquid limit/%	27.2
Plastic limit/%	17.2
Plasticity index	10.0
Optimum water content/%	33.0
Maximum dry density/$g\cdot {\mathrm{cm}}^{-3}$	1.47
pH value	7.88–8.21

**Table 2 materials-14-06985-t002:** Chemical compositions of major materials.

Constituent	Cement/%	Fly Ash/%	Carbide Slag/%
SiO_2_	20.96	50.10	3.0
Al_2_O_3_	4.98	26.50	2.5
CaO	64.03	4.10	59.5
Fe_2_O_3_	3.22	8.40	0.9
Na_2_O	0.07	7.15	—
TiO_2_	—	—	0.77
MgO	1.30	0.85	0.2
SO_3_	2.60	1.40	0.89
K_2_O	0.55	1.50	0.03

**Table 3 materials-14-06985-t003:** Experimental design.

W-d Cycles	Curing Time/d	${\mathrm{Mn}}^{2+}$ Concentration/ $\mathrm{mg}\cdot L^{-1}$	Test Procedures (with Equipment Type)
0,1,3,5,7,10	28	1000	Mass loss—Electronic scales (LQ-C20002) and electric constant temperature blast drying oven (DHG-9246A, JiangSu, China) UCS—Electronic universal testing machine (YSH-229WJ-50kN, ShangHai, China) Resistivity-Digital Bridge (TH2828A, ChangZhou, China) Leaching concentration—Inductively coupled plasma emission spectrometer (Spetro Arcos, Kleve, Germany) SEM and EDS—Electron microscope (UltimaⅣ 2036E102, Tokyo, Japan) XRD—Ultima IV diffractometer (Nippon Rigaku, Tokyo, Japan)
		5000	
	60	1000	
		5000	

**Table 4 materials-14-06985-t004:** Fitting relationship between UCS and IR.

Curing Time/d	${\mathrm{Mn}}^{2+}$ Concentration/$\mathrm{mg}\cdot L^{-1}$	Fitting Formula	Decision Coefficient $R^{2}$
28	1000	$q_{u}=0.010\rho_{0}+1.740$	0.978
	5000	$q_{u}=0.012\rho_{0}+2.378$	0.986
60	1000	$q_{u}=0.011\rho_{0}+2.546$	0.947
	5000	$q_{u}=0.015\rho_{0}+1.106$	0.920

Note: $q_{u}$ is the UCS of the specimen and $\rho_{0}$ is the IR of the corresponding specimen.

**Table 5 materials-14-06985-t005:** Porosity ratio of specimens under different durability conditions at the age of 28 d.

Curing Time /d	Magnifying Power	Contaminant Concentration/$\mathrm{mg}\cdot {\mathrm{kg}}^{-1}$	Cycle Times	Pore Area/${\mu m}^{2}$	Soil Area /${\mu m}^{2}$	Porosity Ratio	Growth Ratio /%
28	×5000	1000	0	166,358	1,024,066	0.162	37.42
			10	209,585	938,853	0.223	
		5000	0	135,532	1,087,255	0.125	28.11
			10	161,337	1,010,247	0.160	

## Data Availability

Data presented in this study are available on request from the corresponding authors.

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
