# Peer review of "Study on the Mechanical and Leaching Characteristics of Permeable Reactive Barrier Waste Solidified by Cement-Based Materials"

_materials, 2021, doi:10.3390/ma14226985_

Round 1
Reviewer 1 Report
Minor comments:
- Authors have provided a literature review of the related studies; however, they have not highlighted the novelty of their study compare to previous studies in the introduction. What research gap has been addressed?
- For the data presented, is there any replications conducted on the samples? If so, it needs to be indicated and error bars need to be added.
- The error bars need to be added to Figure 9 part b.
- Figure 1, the font size of (a) is different than (b), etc.
In this research, the effect of wet-dry cycles on the properties of the solidified permeable reactive barrier (PRB) waste was investigated. The authors monitored mass loss, unconfined compressive strength, resistivity, and Mn2+ concentration under each wet-dry cycle. Their study seems not to be novel compared to previous studies, the only difference seems to be the material used. I added some related studies to the comments provided below.
Additionally, this study by including analytical techniques is trying to correlate the observed trend in w-d cycles with the material properties. However, previous studies such as “Wang, Dongxing, et al. "Strength, durability and microstructure of granulated blast furnace slag-modified magnesium oxychloride cement solidified waste sludge." Journal of Cleaner Production 292 (2021): 126072.” also correlate the change in the mechanical and brucite and C–S–H gels formation. The conclusions are consistent with the result and discussion. However, the discussion from the related study can improve the work.
Summary of the study:
In this research, the effect of wet-dry cycles on the properties of the solidified permeable reactive barrier (PRB) waste was investigated. The authors monitored mass loss, unconfined compressive strength, resistivity, and Mn2+ concentration under each wet-dry cycle. Their results suggest the existence of an optimum number of wet-dry cycles. Additionally, by using microscopy and analytical techniques such as XRD they presume the reason for the observed behavior is attributed to the hydration products. Overall the finding of this study can help to improve the design of the corresponding materials.
Major comments:
- The authors have investigated the effect of the wet-dry cycles on the PRB properties and have provided a literature review of the related studies; however, their study seems not to be novel compared to previous studies. I recommend the authors to revise the introduction to highlight any novelty that is being addressed in their study.
Some related references are:
- Cai, Guang-Hua, Song-Yu Liu, and Xu Zheng. "Effects of Drying-Wetting Cycles on Durability of Carbonated Reactive Magnesia-Admixed Clayey Soil." Journal of Materials in Civil Engineering 31.11 (2019): 04019276.
- Du, Yan-Jun, et al. "Durability of reactive magnesia-activated slag-stabilized low plasticity clay subjected to drying–wetting cycle." European Journal of Environmental and Civil Engineering 20.2 (2016): 215-230.
- Hoy, Menglim, et al. "Effect of wetting–drying cycles on compressive strength and microstructure of recycled asphalt pavement–Fly ash geopolymer." Construction and Building Materials 144 (2017): 624-634.
- Cai, Guang-Hua, Song-Yu Liu, and Xu Zheng. "Influence of drying-wetting cycles on engineering properties of carbonated silt admixed with reactive MgO." Construction and Building Materials 204 (2019): 84-93.
- The authors need to compare the contradiction/similarity between the findings of the related studies. This study by including analytical techniques is trying to correlate the observed trend in w-d cycles with the material properties. However, previous studies such as “Wang, Dongxing, et al. "Strength, durability and microstructure of granulated blast furnace slag-modified magnesium oxychloride cement solidified waste sludge." Journal of Cleaner Production292 (2021): 126072.” also correlate the change in the mechanical and brucite and C–S–H gels formation. Related studies need to be cited.
- In the testing procedure, for the w-d cycle test, the authors have not indicated if the experiment were conducted in replicates. If so, the method needs to be revised and error bars need to be added to corresponding Figures.
- For the results presented in table 3, the confidence interval for the slope and the intercepts needs to be calculated and be provided. The intervals need to be considered for comparison between the slopes and establishing any correlation.
Minor comments:
- The error bars need to be added to Figure 9 part b.
- Figure 1, the font size of (a) is different than (b), etc.
- Figure 9, the format of the y-axis are not consistent and the x-axis needs to be plotted in the same range.
Author Response
Dear Reviewers:
Thank you very much for your review of my article and for your very detailed and pertinent suggestions. I have now made the appropriate changes in the text based on your suggestions, which I will list below:
- Regarding your question about the innovativeness of the study, I have made corrections in the paper.
There has been a great deal of previous research on various solid wastes and the corresponding curing materials, including their properties. In this paper, based on the research on curing technology, the idea of curing treatment is proposed for PRB materials that have reached their service life and are cured using CFC composites, based on which the change in their properties under the influence of wet and dry cycles is studied. both CFC materials and PRB wastes contain different degrees of environmental impact, so the research in this paper is also very beneficial for the sustainable use of the environment.
I would like to thank you for your suggestions of several references that have helped me greatly in revising my article.
- I have also made timely corrections to the flaws in the analysis of the mechanism section in the text.
Based on the original analysis and your comments, I added a section combining micro-analysis and macro-analysis to make the logic and overall framework of the article more complete.
- On the issue of test parallel samples, I have made the corresponding explanations in the text.
In my test process, each group of specimens have 3 parallel samples for control reference, due to my negligence this is not indicated in my graph, have taken your suggestion to make corrections.
- I have also corrected and revised the figure in the article for the issue of adding confidence intervals for the UCS and IR relationship fitting, and your suggestion does make the line content more rigorous.
- I have made corrections to the formatting issues in your suggestions #5-7.
The corrections can be viewed by marking the content, so please check the corrected content. Thank you again for your help with the content of the article, and I look forward to your reply.
yours sincerely,
chenxun

Reviewer 2 Report
The presented article is devoted to the study of important characteristics of PRB waste using modern methods of analysis. While reading the article I have a positive opinion about it. This research refers to the solution of urgent questions of science. In general, the results are well presented. There are a few comments and questions for the authors:
1. The introduction needs to be corrected. You write about the research of different scientists without leading to the logical conclusions that you need to study this as well.
2. Section Materials needs to be corrected. This particular section is very hard to read. How about moving Tables 1 and 2 to Supplementary Materials?
3. For all equipment used it is necessary to give not only the model, but also the company-manufacturer and the country.
4. What is the purity of manganese sulfate and where was it purchased? The text should justify why manganese was chosen to create a model acidic mine drainage.
5. Line 78 - correct to 7:3.
6. Line 117 - why do you write PRB here and not RLC?
7. Line 145-148 - Sentences should be combined.
8. All pictures need to be moved to the place after the paragraph where it is first mentioned.
9. There is no mention of table 3 in the text.
10. The required style should be applied to the list of references.
Author Response
Dear Reviewers:
Thank you very much for your review of my article and for your very detailed and pertinent suggestions. I have now made the appropriate changes in the text based on your suggestions, which I will list below:
- Regarding your question about the innovativeness of the study, I have made corrections in the paper.
There has been a great deal of previous research on various solid wastes and the corresponding curing materials, including their properties. In this paper, based on the research on curing technology, the idea of curing treatment is proposed for PRB materials that have reached their service life and are cured using CFC composites, based on which the change in their properties under the influence of wet and dry cycles is studied. both CFC materials and PRB wastes contain different degrees of environmental impact, so the research in this paper is also very beneficial for the sustainable use of the environment.
- With regard to your questions in Tables 1 and 2, I have adjusted them in the article and explained them accordingly.
- Your comments 3-4 in the instrumentation model, Mn procurement place of the problem, I have added in the article to explain.
- For the formatting issues in your comments 5-9, I have made corrections respectively.
Thank you very much for your very pertinent suggestions, which have greatly helped the completeness and rigor of my article, and I would like to thank you from the bottom of my heart.
Corrections can be viewed by marking the content, so please check the corrected content. We look forward to your reply.
Yours sincerely,
chenxuan

Reviewer 3 Report
Some grammatical errors exist. So, it is recommended to double-check the manuscript by an English editor.
The studied manganese contaminated PRB waste is not suitably described and characterized. So, it is suggested to, first, describe different available types, the adopted one, and then, to go through the in-depth representation of its features.
Reference number 26 is not accessible. As it contains some fundamental parameters and descriptions highlighted in the current study (as referred to in the manuscript), it is suggested to add the information not directly presented in the manuscript instead of referring to that work to fully address the specifications of the adopted material.
To fully characterize the Leaching characteristics of stabilized PRB, permeability features play an important role. It is suggested to provide a comparison between permeability characteristics of untreated and stabilized samples.
It is suggested to define the following ratios (Si/Al, Na/Al, Mn/Al, and Ca/Si) in samples containing different solutions and interpret the corresponding test results with regards to their specific ionic ratios and SEMs.
Curing conditions are not presented in the manuscript. As the curing temperature is an important factor in the formation of the hydration products and strengthening the samples, authors need to explain the curing temperature and conditions. Were the samples cured in the ambient or in an elevated temperature?
The material percentages used to make PRD are not properly described. Why is such a mix-design used (why is only one fixed percentage ratio of CFC used for making the stabilized PRB samples)? The experimental program and the reason behind such a selection (both of selected binder types and percentages) should be clearly presented. How can someone make a justification on the role of each of the used stabilizing agents in strengthening the samples based on one fixed weight percentage?
Standards used to prepare samples (UCS and SEM samples) and to conduct all the tests (compaction test, hydrostatic method, SEM, XRD, etc.) should be clearly mentioned.
It is strongly recommended to add a table to clearly explain the program of the experiment. Weight percentages of used different materials, molarities, type of solution, curing conditions (days and temperatures), conducted tests, and other independent parameters (e.g., ionic ratios, etc.) affecting the test results should be addressed throughout the table.
All used abbreviations should be described at their first appearances. Device models should also be completely presented throughout providing the full name, manufacturer company, and country (e.g., WDW-100, etc.).
It will be beneficial if authors can suggest an empirical correlation for UCS of their studied samples based on concerning parameters and other test results (if there is enough data, and an acceptable correlation of the determination value). Inserting curing time (and temperature if available), Mn concentration, and ionic ratios into the predictive correlation can improve the comprehensiveness of the proposed relationship. Weight percentages of the adopted binders used to stabilize samples could be another interesting input parameter for increasing the comprehensiveness and applicability of the suggested empirical relationship. It can be carried out in each of the multiple-variable regression frameworks.
Author Response
Dear Reviewers:
Thank you very much for your review of my article and for your very detailed and pertinent suggestions. I have now made the appropriate changes in the text based on your suggestions, which I will list below:
- In response to your question about the defective PRB waste description, I have made additions and explanations in the corresponding part of the article.
- For the issue of reference 26, several corrections have been made to the missing parts of the article, and if you need to consult this literature, you can do so in the annex.
- You suggested that the content of the penetration experiment needs to be added. My opinion on this is as follows: My idea is that the whole article is focused on the study of durability, so the content of the penetration section is not listed.
The experiment shows that the cured RLC permeability coefficient is smaller than the uncured RLC, indicating that CFC curing agent can fully meet the permeation requirements. The results of the graphs are shown in the Appendix.
- For EDS elemental analysis has been added, and the combination of EDS and SEM for more detailed analysis of the microscopic specimens.
- The curing conditions of the specimens are described in the text with additions.
- For CFC curing agent ratio selection specific 7:7:5, the following explanation is given.
The ratio was chosen in the article because a specific orthogonal test analysis was conducted on the selection of the ratio during the preliminary tests, and the results showed that the UCS and permeation properties of the specimens achieved the best results with this ratio.
The table of orthogonal test results is attached.
- The experimental specifications, instrumentation models and experimental designs in the recommendations of Articles 7-9 are added and supplemented in the article.
- The last suggestion you made, the idea of refining the various parameters obtained from the experiments in the article into empirical formulas, is indeed a very good idea, but due to the lack of ability and various conditions at present, I may not be able to meet such requirements, but I will certainly approach this aspect in my future research, so that my research can also be more widely used in practice.
- Sorry, I can't send the chart in this way, I have sent it to the editor's mailbox, please check through the editor.
Thank you very much for your very pertinent suggestions, which have greatly helped the completeness and rigor of my article, and I would like to thank you from the bottom of my heart.
Corrections can be viewed by marking the content, so please check the corrected content. We look forward to your reply.
Yours Sincerely,
chenxuan
Round 2
Reviewer 3 Report
Based on the made corrections, the manuscript is acceptable for publication.